The diet of otters (Lutra lutra) on the Agri river system, one of the most important presence sites in Italy: a molecular approach

Buglione Maria 1
Petrelli Simona 1
Troiano Claudia 2
Notomista Tommaso 1
Rivieccio Eleonora 1
Fulgione Domenico fulgione@unina.it 1
1 Department of Biology, University of Naples Federico II , Naples , Italy , Italy
2 Department of Humanities, University of Naples Federico II , Naples , Italy , Italy
Yoccoz Nigel
Electronic publication date: 2020 Jul 29
Publication date: 2020
Volume: 8
Electronic Location ID: e9606
Received 2020 Mar 26; Accepted 2020 Jul 5
Copyright: ©2020 Buglione et al.
Copyright year: 2020
Copyright holder: Buglione et al.
License: This is an open access article distributed under the terms of the Creative Commons Attribution License, which permits unrestricted use, distribution, reproduction and adaptation in any medium and for any purpose provided that it is properly attributed. For attribution, the original author(s), title, publication source (PeerJ) and either DOI or URL of the article must be cited.
License URL: https://creativecommons.org/licenses/by/4.0/

Keywords: Otter, DNA metabarcoding, Fish, Food ecology, Conservation

Funding: The authors received no funding for this work.

==============================
Background

The Eurasian otter (Lutra lutra) underwent a strong decline in large areas of the Central-Western part of its distribution range, during the second half of the twentieth century. In Italy, only residual fragmented nuclei survive in the Central-Southern part of the peninsula. Nowadays, the otter is one of the most endangered mammals in Italy, and increasing the knowledge about the ecology of this species is a key step in defining fitting management strategies. Here we provide information about the diet of otter on the Agri river system, one of the most important presence sites in Italy, to understand both the species’ food requirements and the impact on fish communities.

Methods

DNA metabarcoding and High Throughput Sequencing were used on DNA extracted from spraints. We amplified DNA with a primer set for vertebrates, focusing efforts on the bulk of the otter’s diet (fishes and amphibians).

Results

Our findings showed that the diet of the otter was dominated by cyprinids (97.77%, and 99.14% of fishes), while amphibians represented 0.85% of the sequences analyzed. Results are in general accordance with previous studies based on morphological characterization; however, molecular analyses allow the resolving of some morphological uncertainties. Although the study area offers a very wide range of available prey, the diet of the otters shows marked selectivity. We highlighted a variation in prey consumed, in accordance with the typology of water system (i.e., river, lake, tributary). Some of the preys found in the diet were alien species introduced by man for sport fishing. Our findings could help define strategies useful for the conservation of the otter population in Southern Italy, suggesting management actions directed at avoiding fish community alterations through illegal stockings without severe controls on their taxonomic status. These introductions could result in a general reduction in the diversity of the otter’s preys, affecting its predatory behavior.

Introduction

The Eurasian otter (Lutra lutra) is a semi-aquatic carnivore widespread in freshwater habitats of Europe and Asia. During 1960–1968, the species underwent a strong decline in large areas of the Central-Western part of its distribution range (Mason & Macdonald, 1986; Mason & Macdonald, 1994; Conroy & Chanin, 2000; Hilton-Taylor, 2000; Conroy & Chanin, 2002) mainly due to human persecution, persistent organic pollutants, loss of riparian habitats (Mason, 1995; Murk et al., 1998; Wren, 2001; Ruiz-Olmo et al., 2002; Panzacchi, Genovesi & Loy, 2010) and fish community alterations as a result of over-fishing (Arcà & Cassola, 1986; Carss, 1995).

In Italy, this severe decrease led to the total disappearance of the otter in the Northern part of the peninsula (Macdonald & Mason, 1983; Cassola, 1986; MacDonald & Mason, 1994; Prigioni, Balestrieri & Remonti, 2007) and to a gradual decline in Central and Southern Italy (Spagnesi, 2002; Boitani, Lovari & Vigna Taglianti, 2003; Prigioni, Balestrieri & Remonti, 2007), with residual fragmented populations mainly in Basilicata, Campania and Calabria regions (Prigioni, 1997).

At present, the Southern-Italian population, geographically and genetically isolated from all otter populations living in Europe (Spagnesi, 2002; Randi et al., 2003), occurs in a continuous range from Abruzzo to Basilicata, Calabria and Apulia (Giovacchini et al., 2018). Despite a general positive trend and a slow recovery with an expansion northward (Fusillo, Marcelli & Boitani, 2003; De Castro & Loy, 2007; Prigioni, Balestrieri & Remonti, 2007; Balestrieri et al., 2008; Loy et al., 2009; Buglione et al., in press), the otter in Italy is one of the most threatened mammals, listed as Endangered (EN, category D) on the Official Italian Red List (Loy et al., 2010; Rondinini et al., 2013; Scorpio et al., 2014), both at national and local levels.

In this scenario, increasing the knowledge about the ecology of the species is a key step in understanding factors impacting this population, in order to better define fit management strategies.

We focused our study on a very important presence site of the otter in Italy, the Agri river system (in the Basilicata region), which flows through the internal part of Southern Italy up to the Ionian Sea.

This river system hosts one of the main, stable otter populations (Prigioni et al., 2006b), and its upper course represents a precious connection between the otter populations of different regions in Southern Italy (Campania, Basilicata and Calabria) (Smiroldo et al., 2009).

The Basilicata region accounts, together with the Campania region, for more than 70% of the species’ range (Prigioni et al., 2005), which spans an area between Naples and the Ionian Sea. The latest survey reports an estimated otter population for the entire Agri basin (237 km) of 43–47 individuals (Prigioni et al., 2006b). Given the cumulative length of the watercourses we monitored (136 km), there are approximately 25 individuals in the study area.

Here, we provide data on otter diet diversity for the population of the Agri system, obtained via DNA metabarcoding and High Throughput Sequencing (HTS) analysis on DNA extracted from spraints.

We collected spraints in five sample sites which included both the main river and its tributaries, as well as an artificial lake. All these represent diverse freshwater habitats which could influence the composition of the otter’s diet differently.

Moreover, exploring the putative variation, in light of food availability, could reveal taxonomic entities (among the preys) that are difficult to collect with direct methods focusing on fish communities. This information is of great importance, mainly considering the current habitat modifications and ecosystem alterations that might induce unpredictable changes in abundance, diversity, or availability of resources relevant for this animal (Sanderson et al., 2002; Thuiller et al., 2011). The impact of new alien species could be significant (Prigioni et al., 2006a), so investigating the role of native versus exotic species in the otter’s diet could also provide useful information about possible alterations of predatory behavior.

Material and Methods

Study area

The study was performed on the upper course of the Agri river flowing in the Basilicata region (40°17′N– 15°58′E, Southern Italy; Fig. 1).

Figure 1 Study area.

Location of collected samples (red spots), grouped in five pools. The blue lines highlight the main rivers (order 1) and their tributaries (order 2, 3 and 4 according to waterway hierarchy). The dashed red lines represent regional boundaries. In the inset, the current otter distribution (inferred from Loy et al. (2015) is reported in orange and the study area is defined by the black rectangle.

The Agri river’s source is located on Monte Maruggio, Lucan Apennines at the North of Monte Volturino. Along its course, the Agri river is enriched by various tributaries. Torrential flow characterizes most of its watercourses, which are forced to reduce their speed at downstream reaches, where the Agri river is dammed, creating the Pietra del Pertusillo lake. This artificial lake is located on Grumento Nova, Montemurro e Spinoso territories and consists of an area of 7.5 km2 with a maximum capacity of 150 million cubic meters of water. The entire area falls within the Appennino Lucano and Val d’Agri-Lagonegrese National Park. Proceeding toward the East, the Agri river further widens its riverbed, flowing in a pattern of intertwined canals, typical of a Fiumara of Southern Italy. Finally, it flows into the Ionian Sea with a small estuary (Gulf of Taranto).

The study area covers a land in which rivers cross very different environments in a state of brook, mature river and even lake. This environmental diversity greatly affects the availability of prey for the otter, which, according to current literature, could feed on aquatic animals, such as fishes, amphibians and crustaceans, as well as terrestrial animals, such as reptiles, or on remains of mammals and birds (Clavero, Prenda & Delibes, 2003).

The fish assemblages of the Agri basin are mainly those of the Central Peri-Mediterranean region (BR2 in Reyjol et al., 2008), dominated by cyprinids and trout. More in detail, the recent Fish Plan Management of the Basilicata region (Caricato, Canitano & Montemurro, 2014) drafted the regional fish map highlighting the presence of endemisms, such as Alburnus alburnus, Barbus plebejus, Squalius squalus, Tinca tinca, and allochthonous species from other Italian regions (i.e., Alburnus alborella, Salmorutilus rubilio, Scardinius erythrophthalmus, Salmo trutta) and Asiatic regions (i.e., Carassius carassius, Carassius auratus, Cyprinus carpio, Esox lucius, Rutilus rubilio) (Prigioni et al., 2006a; Caricato, Canitano & Montemurro, 2014).

Furthermore, quantitative information about fish assemblages in the Agri river provided by an electro-fishing census (Prigioni et al., 2006a), showed that, considering a total of 180 fish caught, Barbus plebejus, Barbus meridionalis (30.0%) and Rutilus rubilio (27.8%) were the most abounded fish species, followed by Alburnus arborella and Alburnus albidus (12.8%), Salmo trutta (11.1%), Micropterus salmoides (11.1%), Leuciscus cephalus (5.6%) and Lepomis gibbosus (1.6%).

Some introduced fishes (i.e., Micropterus salmoides and Lepomis gibbosus) replaced the native fauna at the confluences of the Agri and Maglie rivers with the Pietra del Pertusillo lake, in which these exotic fish species are very common (Prigioni et al., 2006a).

Among amphibians, reported species are Bombina pachypus, Bufo bufo, Bufo balearicus, Rana italica, Pelophylax synkl.hispanicus and Lissotriton italicus (Romano et al., 2012).

Sample collection

Sampling was performed from May to December 2018, authorized by Ente Parco Nazionale dell’Appennino Lucano, Val d’Agri-Lagonegrese. A team of collectors walked along transects of approximately 5 km, checking both riversides of the watercourses, rocks and structures emerging from the water, searching for otter scats (spraints).

Field activities determine the success of genetic analysis. In fact, old samples or bad preservation affect the degradation of nucleic acids (Taberlet et al., 1996; Dallas et al., 2000; Goossens et al., 2000; Jansman, Chanin & Dallas, 2001). Therefore, we only collected fresh spraints (N = 51, scats < 2 day old), the freshness of which was determined by skilled field operators using odor and aspect patterns. Old spraints were removed the day before each sampling session, with collection taking place the following day in the early morning, to reduce the time interval from defecation to collection. Furthermore, all scats were manipulated with sterilized tools and the samples were placed in a sterile tube together with silica granules to speed up drying (Wasser et al., 1997; Buglione et al., 2020b).

Each record was geo-referenced using a global positioning system (GPS; UTM-WSG 84) and loaded in a GIS environment using QGIS 3.4.2 (Fig. 1).

Diet analysis

DNA extraction

In order to prevent potential contamination, all genetic analyses were performed in a dedicated laboratory used exclusively for environmental DNA processing. Total genomic DNA was extracted from spraints using QIAamp DNA Fast Stool Mini Kit (QIAGEN, Valencia, CA), following manufacturer’s guidelines. All experiments were performed including a negative control to check for potential cross-contaminations. The performance of the process was evaluated using 1% agarose gel electrophoresis in buffer TBE 1X, while purity and concentration of extracted DNA were checked using a Nanodrop ND-2000 (Nanodrop, Wilmington, DE, USA) and a Qubit Fluorometer 3.0 (Invitrogen by Thermo Fisher Scientific), respectively.

Selection of the molecular marker

According to previous studies that analyzed the diet of the otter using a morphological approach, fishes and amphibians formed the bulk of the otter’s diet (Prigioni et al., 2006a; Remonti et al., 2008; Balestrieri, Remonti & Prigioni, 2009; Smiroldo et al., 2009). Starting from this consideration, we decided to use a primer set for vertebrates.

Therefore, polymerase chain reaction (PCR) amplifications were performed using 16Smam_1 (5′-CGGTTGGGGTGACCTCGGA-3′) and 16Smam_2 (5′-GCTGTTATCCCTAGGGTAACT-3′) primers, that allow the amplification of a small DNA fragment of about 140 bp of 16S rRNA mitochondrial (mt) gene in all vertebrates (Taylor, 1996; Ficetola et al., 2010), useful when working on highly fragmented and heterogenous materials such as faecal DNA (Valentini, Pompanon & Taberlet, 2009; Vynne et al., 2012).

We assessed the ability of the primer set to bind and amplify the DNA by performing multiple alignments of the primer sequences and 16S mtDNA of the otter’s potential prey in the study area (Caricato, Canitano & Montemurro, 2014), downloaded from National Center for Biotechnology Information (NCBI GenBank) nucleotide database (Table S1). Then, we correlated the number of mismatches between primer and each sequence, as this could affect amplification success, misrepresenting the true diversity (Geisen et al., 2015).

Amplification of 16S rRNA gene

The amplification primers were modified with an Illumina adapter added to 5′ends (Valentini, Pompanon & Taberlet, 2009; Coissac, Riaz & Puillandre, 2012).

Each tube contained a total of 25 µL of PCR mix composed by 50 ng of DNA, Taq polymerase, 0.8 M Tris-HCl, 0.2 µM (NH4)2 SO4, 0.2% w/v Tween-20, 2.5 mM MgCl2, 0.4 mM dNTP mix and 0.2 µM of each primer. For each sample, two PCR replicates were performed. Furthermore, a negative control (sterilized distilled water) was included during this procedure to check the performance of the reactions.

The cycles of amplification started with 94 °C for 5 min and 35 cycles of 95 °C for 1 min, 55 °C for 1 min, 72 °C for 90 s, and a final extension step at 72 °C for 5 min.

Pooling of samples

After purification with Illustra GFX PCR DNA and Gel Band Purification Kit (GE Healthcare, Buckinghamshire, UK), PCR replicates for each sample were combined, and DNA in all PCR samples was mixed in equimolar concentrations (Harris et al., 2010) to obtain 5 pools, according to the geographical distribution of the spraints. Each pool was created as a representative of a different type of watercourse, assuming that the content was representative of the prey spectrum of a specific type of river/lake:

Pool 1: The Verzarulo stream, tributary located near the origin of the Agri river (N = 8);

Pool 2: main course of the Agri river, located upstream of the Pietra del Pertusillo lake, at the confluence with the Caolo river (N = 12);

Pool 3: the Maglie river, flowing into the Pietra del Pertusillo lake (N = 9);

Pool 4: the Pietra del Pertusillo lake (N = 12);

Pool 5: main course of the Agri river flowing downstream of the Pietra del Pertusillo lake, at the confluence with the Racanello river (N = 8).

High Throughput Sequencing and bioinformatic analysis of sequence data

Large-scale sequencing was performed with a 2 ×150 bp paired-end run using the Illumina MiSeq platform (Illumina, Inc., San Diego, CA, USA), chosen because of lower error rates than other next generation platforms (D’Amore et al., 2016).

The Nextera DNA Sample Library Preparation protocol was performed at Genomix4Life Srl (http://www.genomix4life.com/it/). The negative control (sterilized distilled water) and an internal positive control were processed together with our samples in high throughput sequencing. Additionally, a sample containing a known microorganism was included in each sequencing run on the TapeStation (Agilent Technologies, Santa Clara, CA).

Analysis of sequencing data, using software (Camacho et al., 2009; Luo et al., 2012; Bokulich et al., 2013; Bolger, Lohse & Usadel, 2014) implemented in Linux environment, was conducted as previously described in (Buglione et al., 2018), with slight modifications according to the molecular markers utilized. In particular, we trimmed and then cropped the reads with low phred quality score (Q < 30) and a minimum length of 35 nt. During the blast of the contig sequences against the nucleotide records in NCBI (National Center for Biotechnology Information), we only selected alignments with an E-value < 0.05, a ratio between the length of the sequence and the alignment (alignment score) > 80% and identity > 90%. Furthermore, the blast results were filtered considering a list of otter prey availability to increase the accuracy of automatic taxonomic assignation (Caricato, Canitano & Montemurro, 2014).

The software bwa-0.7.12 (Li & Durbin, 2009), samtools 1.3 (Li et al., 2009) and samstat 1.5.1 (Lassmann, Hayashizaki & Daub, 2011) were used to perform mapping for quantitative analysis. The reference contigs, assigned to the corresponding taxa during the blast, were used for the alignment of reads. Both chimeric alignments (supplementary alignments) and secondary alignments were detected and removed with samtools 1.3 (see SAM Alignment/Map Format Specification at https://github.com/samtools/hts-specs). Finally, in the calculation of the number of reads for a taxonomic assignment, we considered all sequences with count > 1 (Mollot et al., 2014).

Statistical analysis

For each pool, we calculated the percent of occurrence (PO%) of each taxon as the number of the DNA reads assigned to a taxon divided by total number of DNA reads of all identified taxa in a pool. Furthermore, we provided the relative read abundance (RRA) of the taxa in the otter’s diet, defining it as the average of the percent of occurrence of a specific taxon across all samples. This result was shown in a box plot reporting minimum, maximum, median, quartiles, upper and lower whiskers, and outliers.

Alpha diversity descriptors (richness and the Shannon Index) were elaborated for the five pools using Past v. 3.2. software (Hammer, Harper & Ryan, 2001) and Pearson’s chi-square test was applied to data sets to assess how likely any observed differences between pools were due to chance. Finally, to examine the variation of preys between the pools and highlight the taxa most represented in the five different sampling sites, we performed a Correspondence Analysis (CA) implemented in R v. 3.6.1 (Borcard, Gillet & Legendre, 2011).

Results

Considering all the collected samples, DNA was successfully extracted in 96.07% of cases (49 samples) with DNA concentration ranging between [20–885 ng/µL] and with both λ260∕280 and λ260∕230 > 1.50.

The primers bound the DNA of all the considered potential prey in the study area (fishes and amphibians) with a mismatch between primer and prey sequence not higher than 4 (forward) or 1 (reverse) nucleotide bases (Fig. S1).

A total of 20,957,120 short raw reads (150 bp) was obtained from Illumina MiSeq sequencing of the DNA extracted and amplified from the 49 spraints. During library preparation, no amplicons were detected for the negative control and, similarly, no reads derived from its sequencing on the Illumina platform, revealing no out contaminations.

After trimming, 6,717,838 filtered reads were retained for subsequent analyses. After assembly, 123,500 sequences were used for the blasting against NCBI database, of which 96.54% found a correspondence with deposited nucleotide sequences. Finally, we obtained 113,296 sequences on which we performed quantitative analysis, calculating the number of reads for each taxonomic assignment. Bioinformatics processing of the sequencing data was performed separately for the 5 pools (see details in Table 1).

Table 1 Data processing.

Details of bioinformatic data processing from High Throughput Sequencing on Illumina MiSeq platform of DNA extracted from otter spraints. Quality control (QC); read Forward (read F); Read reverse (read R); Chromosome (Chr); map quality (mapQ).

in total (QC-passed reads + QC-failed reads)	2503050 + 0	700288 + 0	1146568 + 0	716801 + 0	1931112 + 0	
secondary	0 + 0	0 + 0	0 + 0	0 + 0	0 + 0	
supplementary	15616 + 0	12194 + 0	90332 + 0	148249 + 0	13590 + 0	
duplicates	0 + 0	0 + 0	0 + 0	0 + 0	0 + 0	
mapped	2482444 + 0 (99.18%: N/A)	683815 + 0 (97.65%: N/A)	1134865 + 0 (98.98%: N/A)	702198 + 0 (97.96%: N/A)	1905046 + 0 (98.65%: N/A)	
paired in sequencing	2487434 + 0	688094 + 0	1056236 + 0	568552 + 0	1917522 + 0	
read F	1243717 + 0	344047 + 0	528118 + 0	284276 + 0	958761 + 0	
read R	1243717 + 0	344047 + 0	528118 + 0	284276 + 0	958761 + 0	
properly paired	2180922 + 0 (87.68%: N/A)	294 + 0 (0.04%: N/A)	39188 + 0 (3.71%: N/A)	23656 + 0 (4.16%: N/A)	530 + 0 (0.03%: N/A)	
with itself and mate mapped	2446254 + 0	655200 + 0	1032912 + 0	539538 + 0	1865562 + 0	
singletons	20574 + 0 (0.83%: N/A)	16421 + 0 (2.39%: N/A)	11621 + 0 (1.10%: N/A)	14411 + 0 (2.53%: N/A)	25894 + 0 (1.35%: N/A)	
with mate mapped to a different chr	265330 + 0	556252 + 0	931324 + 0	445678 + 0	1732242 + 0	
with mate mapped to a different chr (mapQ>=5)	49779 + 0	28183 + 0	40633 + 0	42070 + 0	36160 + 0	

The automatic taxonomic assignment of the sequences revealed the presence of DNA ascribable to mammals (otters, wild boars, hares, cows, goats, rats and mice) and birds, although represented by few sequences.

These findings suggest an interaction with these species or with parts of them (such as excrements, carcasses or fur), although this does not necessarily imply predation events (O’Sullivan, Sleeman & Murphy, 1992). For this reason, these sequences, as well as otter DNA sequences, were not considered in the final processing of the diet data, retrieving only sequences assigned to Pisces and Amphibia.

The results from molecular analyses of the otter’s diet showed that the total number of reads was dominated by DNA from fishes (99.14%) while amphibians represented 0.85%, mirroring an estimate of the relative proportion of these items ingested by the otter (Deagle, Kirkwood & Jarman, 2009; Pompanon et al., 2012), and resulted in 6 families (Fig. 2 and Table 2).

Figure 2 Diet of all otter samples analyzed.

(A) Number of reads count (log 10), from all five pools, assigned to each family. Blue, Cyprinidae; pink, Percidae; green, Ranidae; orange, Salmonidae; red, Bufonidae; yellow, Gobidae. (B) Box-plot showing the Relative Read Abundance (log 10) of the most represented fish and amphibian taxa revealed in the diet of otters. The histogram represents the interquartile range with the median (bold black line), the outliers (empty circles), the whiskers (dashed vertical lines) and the minimum and maximum (horizontal lines). Blue, Cyprinidae; pink, Percidae; green, Ranidae; orange, Salmonidae.

Table 2 Details of the otter’s diet.

Qualitative (Family and genus/species) and quantitative (Percent of Occurrence %) analysis of otter diet.

	Family	Genus/Species	Percent of occurrence (%)	
			Pool 1	Pool 2	Pool 3	Pool 4	Pool 5	
	Cyprinidae	Carassius sp.	1.9386	25.6767	62.0621	39.7780	0.4517	
	Cyprinidae	Barbus plebejus	34.0436	8.4140	5.0461	8.2649	45.0129	
	Cyprinidae	Cyprinus carpio	0.1538	1.9892	3.4597	2.2342	0.0078	
	Cyprinidae	Alburnus albidus	51.2940	1.5838	0.6100	0.0000	1.9001	
	Cyprinidae	Squalius squalus	0.1319	2.7483	0.3560	1.2348	30.6459	
	Cyprinidae	Sarmarutilus rubilio	0.0148	0.6142	0.0162	3.5938	18.5180	
	Ranidae	Rana italica	0.2424	2.7500	25.1475	0.0000	0.0000	
	Percidae	Perca fluviatilis	0.0000	0.0000	0.0920	44.7242	0.1294	
	Cyprinidae	Rutilus rutilus	0.0139	0.0811	0.0180	0.0709	0.5514	
	Cyprinidae	Scardinius scardafa	0.0019	0.0000	0.0036	0.0000	0.0725	
	Salmonidae	Salmo trutta	0.0000	0.2967	0.7995	0.0000	0.0000	
	Cyprinidae	Barbus barbus	0.0304	0.0000	0.0000	0.0000	0.1514	
	Cyprinidae	Protochondrostoma genei	0.0000	0.0000	0.0000	0.0000	0.0078	
	Cyprinidae	Ballerus sapa	0.0000	0.0000	0.0000	0.0000	0.2459	
	Cyprinidae	Alburnus arborella	0.0000	0.0000	0.0000	0.0000	0.1851	
	Cyprinidae	Luciobarbus sp.	0.0000	0.0000	0.0000	0.0000	0.1450	
	Cyprinidae	Aspius aspius	0.0003	0.0000	0.0000	0.0028	0.0000	
	Gobidae	Padogobius nigricans	0.0000	1.1145	0.0000	0.0000	0.0000	
	Bufonidae	Bufo bufo	0.0000	0.0000	0.0036	0.0000	0.0000	
	Cyprinidae	Blicca bjoerkna	0.0000	0.0000	0.0000	0.0000	0.0026	
	Cyprinidae	unclussified Cyprinidae	12.1350	54.7315	2.3859	0.1248	1.9726	
Total	6	21	100	100	100	100	100	

Cyprinidae (97.77% of total reads) was the dominant family, consisting of 16 different species. The Italian bleak (Alburnus albidus), together with the Italian barbel (Barbus plebejus), accounted for 77.24% of the total reads of DNA sequences for all samples, and 79% of all Cyprinidae species (Fig. 2A and Table 2). The remaining families were each represented by one species: Percidae (the common perch Perca fluviatilis, 1.31%), Ranidae (the Italian stream frog Rana italica, 0.85%), Gobidae (the Arno goby Padogobius nigricans, 0.026%), Salmonidae (the brown trout Salmo trutta, 0.025%) and Bufonidae (the common toad Bufo bufo, < 0.01%) (Fig. 2A and Table 2). The relative read abundance of the different species in the food of the otter (Fig. 2B and Table 2) highlights how the Italian barbel, the Italian bleak, the Italian chub (Squalius squalus), and the South European Roach (Sarmarutilus rubilio) are the most represented indigenous species, even if we cannot reliably conclude that these are the real relative relationships among the preys. The common carp (Cyprinus carpio), the perch, the trout and the frog show appreciable levels of relative read abundance, albeit with a large variance due to their spatial segregation. In particular, carps and perches are present in high percentage in the diet of otters from the Pietra del Pertusillo lake while frogs are included in the diet of otters sampled on the river courses upstream of the lake.

The diet characterization of the spraints in pool 1 revealed 12 taxa with a resolution at species level in 85.92% of cases, at genus level in 1.93% of cases and at family level in 12.13% of cases. Cyprinidae (99.75%) was the most abundant family ingested by the otters, followed by Ranidae (Rana italica, 0.24%). Considering Cyprinidae, the Italian bleak and the Italian barbel seem to be the most abundant fish items, with a percent of occurrence ≥ 30%. We also revealed sequences assigned to the Tyrrhenian rudd (Scardinius scafandra) and the asp (Aspius aspius), although at very low percentages (< 0.002%) (Table 2).

Analysis of DNA extracted from spraints included in pool 2 showed 11 taxa with identification at species level for 19.59% of the sequences, at genus level for 25.67% and at family level in 54.73% of cases. For pool 2, Cyprinidae (95.83%) was again found to be the most occurring family in the diet of otters sampled on the Agri-Caolo watercourse, followed by Ranidae (2.75%), only represented by the Italian frog. The Crucian carp (Carassius sp.) was the most frequently occurring item (25.79%) among Cyprinidae, followed by the Italian barbel (8.77%), the Italian chub (2.86%), the common carp (2.07%) and the Italian bleak (1.65%). All remaining cyprinid species found had a percent of occurrence < 1%. Besides Cyprinidae, Gobidae (1.15%) and Salmonidae (0.29%) families were found. In particular, the brown trout represented the only species revealed for Salmonidae (Table 2).

Spraints in pool 3 showed a diet composed by 13 taxa, assigned at species level, genus level and at family level in 35.52%, 62.06% and 2.38% of cases, respectively.

Cyprinidae (73.95%) was the most frequently occurring family, of which Carassius sp. (83.91%) exhibited the highest percent of occurrence. Cyprinidae was followed by Ranidae (25.14%), Salmonidae (0.80%), Percidae (0.09%) and Bufonidae (0.004%) (Table 2).

Food determination for spraints in pool 4 showed a diet composed by 9 fish taxa, with a percentage of sequence assignment at species level of 60.12%, at genus level of 39.77% and at family level of 0.12%. Diet was dominated by Cyprinidae (55.30%) and Percidae (44.72%). Carassius sp. was the most frequent taxa in Cyprinidae (71.92%), followed by the Italian barbel (14.94%). The remaining sequences had a percent of occurrence < 10% (Table 2).

At last, diet analysis for spraints in pool 5 showed the highest number of taxa (S = 16; Fig. 3) but they are included in only two fish families (Cyprinidae and Percidae), with 97.57% of sequences assigned to species level. For these samples as well, Cyprinidae (99.77%) was the dominant family. Considering the latter, the Italian barbel showed the highest percent of occurrence (45.11%), followed by the Italian chub (30.71%) and the South European Roach (18.55%). The other cyprinid species were found with a percent of occurrence <2%. We revealed, although at low percentages (<0.6%), some fish species typical of Caucasus areas (between the Black Sea and the Caspian Sea), like the white-eye bream (Ballerus sapa, 0.24%) and the white bream (Blicca bjoerkna, 0.002%) (Table 2).

Figure 3 Variation of otter diet diversity and relatedness in the five types of watercourses.

(A) Fish families found in each river type, the pie diagrams show the taxonomic composition of the otters’ diet at family level. For each pool are indicated the richness (S) and Shannon-Weaver Index (H), the differences among indices are significant (Chi-square value: 958.4, degrees of freedom: 40, p < 0.05). Blue, Cyprinidae; pink, Percidae; green, Ranidae; orange, Salmonidae; red, Bufonidae; yellow, Gobidae. (B) Correspondence Analysis (CA) comparing the diet composition of the five pools (object) using fish taxa (variables). Others represent Luciobarbus sp., Ballerus sapa, Blicca bjoerkna, Protochondrostoma genei and Alburnus alborella. In the inset, eigenvalues (% of the total) are reported.

The diet of the otter changes according to the structure of the watercourse (Fig. 3). The greatest diet diversity (meaning richness in fish and amphibian families preyed) characterizes the otters from the main course of the Agri river, downstream of the Pietra del Pertusillo lake (pool 5). In the lake, the number of species in the spraints mainly belongs to Cyprinidae and Percidae, showing a drastic depletion in number of species detected in the scats (pool 4). On the rivers and tributaries, upstream of the lake (pools 1, 2 and 3) spraints are characterized by the presence of items from the Ranidae family, although in different percentages, and the diet richness, in terms of items ingested by otters, is comparable. The highest value of Shannon–Weaver index was recorded on the main river Agri (pool 2) (Fig. 3A).

Correspondence Analysis ordered the diets taken from the pools based on the information deriving from the variables (items). Total variance generated the dimensions DM1, DM2 and DM3 that accounted for 37.07%, 26.34% and 20.75%, respectively. Biplot highlights the contribution of the most significant variables in the relative arrangement among the pools (Fig. 3B). The diet of pool 1 is affected mainly by the presence of the Italian bleak. The Italian barbel, together with the contribution of the Italian chub and the South European Roach, characterize the composition of pool 5, downstream of the lake. The diet of the otters on the Pietra del Pertusillo lake (pool 4) is influenced mainly by the common perch, whereas Carassius sp. affects the diet of the otters on the Maglie river (pool 3). Unclassified cyprinids weigh heavily in the diet of pool 2 (Fig. 3B).

Discussion

The Eurasian otter is a top predator in freshwater habitats, can therefore play an important role in the functioning and structuring of the ecosystem (Ayres & García, 2011; Day, Westover & McMillan, 2015; Berger et al., 2001). The analysis of its food habits could improve the knowledge of both the ecological requirements of this species and of the species composition of communities from the river system.

Generally, all the fish species inhabiting the Agri river and its tributaries reproduce in the period from May until the summer, with a production of juvenile fish in summer and autumn.

This is important because it could influence the rate at which a certain taxon is consumed. Our sample collection covers this period so that the otters we surveyed had the opportunity to prey on individuals of different age, of the species identified.

The ontogenesis of fish species found in the otter’s diet could be very variable, with a potentially wide spectrum of body sizes. However, the size of the river, the salinity and the temperature evens communities and fish body sizes among adults of various species. Our analyses have, most probably, been conducted on otters of different genders and at different life stages. Indeed, working with scats we knew that no gender or age-related restrictions exist in faecal marking activities. Both sexes used the marking points and the high number of droppings from juvenile otters demonstrates that the use of marking points is not restricted to adults (Kalz, Jewgenow & Fickel, 2006).

DNA metabarcoding coupled with HTS represented a good approach for analyzing the diet of otters from Southern Italy using DNA extracted from non-invasive samples, in terms of accuracy, cost, time, and effort (Deagle, Kirkwood & Jarman, 2009; Soininen et al., 2009; Pompanon et al., 2012; Berry et al., 2015; Kumari et al., 2019; Martínez-Abraín et al., 2020).

However, in the interpretation of data, we proceeded with caution. In fact, the use of reads count as a direct measure of ingested biomass remains a highly debated issue (Deagle et al., 2019; Kumari et al., 2019) and some considerations should be taken into account before doing so. Ideally, the number of sequences assigned to a taxon would mirror the biomass in faecal material. However, several factors, acting from collection to sequencing, could alter this correspondence (Deagle, Kirkwood & Jarman, 2009; Vynne et al., 2012; Valentini, Pompanon & Taberlet, 2009; Pompanon et al., 2012; Deiner et al., 2017; Gloor et al., 2017; Porter & Hajibabaei, 2018). Indeed, rather than the absolute interpretation of data, reads count represents a comparative estimation among items revealed by sequencing analysis (Pompanon et al., 2012; Deagle et al., 2019). In our case, taxonomic assignment allowed us to discriminate at species level resolution. In some cases, the incompleteness of the reference database or the quality of the extracted DNA only allowed assignment at the family or genus level. For example, after automatic taxonomic assignment and filtering, we decided to assign 6.08% of the total sequences to the genus Carassius, without attempting a specific diagnosis. Indeed, the characterization of the fish fauna in the Basilicata rivers (Caricato, Canitano & Montemurro, 2014) revealed the presence of Carassius carassius and Carassius auratus.

Our findings showed that the diet of the otter is dominated by cyprinids, in agreement with their abundance in the Southern Italian rivers (Prigioni et al., 2006a; Bianco, 2014).

Although it is necessary to carefully evaluate the comparisons between the percentages of presence of each item sequenced, an autochthonous species was the predominantly consumed species in the case of Alburnus albidus (45.27%) and Barbus plebejus (31.95%).

The allochthonous Carassius sp. and Perca fluviatilis follow in importance as components of otter diet, and they show a wide distribution in pools of our study area. Also, Salmo trutta could refer to the introduced Atlantic lineage, given its appeal to sport fishermen. Moreover, our sequences do not blast against the indigenous Mediterranean lineage deposited in the NCBI database. However, due to taxonomic disagreement on these species in the scientific community, we need further and in-depth analysis to disentangle this doubt.

We only found Salmo trutta in samples collected at the confluence between the Caolo and the Agri river (pool 2) and in samples from the Maglie river (pool 3), together with cyprinids. The latter finding underlines the ability of otters to reveal species that did not emerge during the previous electrofishing survey, which reported trout as the only fish species in the Caolo river and in the upper part of the Maglie river (Prigioni et al., 2006a).

Many results suggest that the otter is a generalist predator, and its diet seems to vary according to prey availability (Prigioni et al., 2006a). In the case of Squalius squalis, reported only in the lowest part of the Agri river (named Leuciscus cephalus in (Prigioni et al., 2006a), our data show its presence in the spraints from all pools in a percentage ranging from 0.13%, in the upper part of Agri to 30.64% in the lower part.

Rutilius rubilio was indicated as one of the preferred food items for otters in Southern Italy (Prigioni, 1997) and a gradual decline in its frequency of occurrence was registered from 1989–1990 (50%) to 2001–2002 (15%), ascribable to an increase in centrarchid consumption (Balestrieri et al., 2006; Prigioni et al., 2009). Our diet analysis revealed Salmorutilius rubilio [synonymous] in all sites, mainly downstream of the dam (18.51%) where it is probably the most abundant species.

In some cases, the molecular method revealed the presence of species that were not previously recorded in the study areas, for example the allochthonous species Aspius aspius, Ballerus sapa, Blicca bjoerkna, Luciobarbus sp., Padogobius nigricans. Moreover, Aspius aspius, introduced in Italy in the twentieth century, is a novelty for the Agri river, and it has been reported only up to central Italy in some artificial basins in the province of Rieti (Bianco, 2014).

These findings probably depend on the higher sensitivity of molecular methods in both food detection and identification, revealing also non-solid items or preys ingested at low percentages (Soininen et al., 2009; Ando et al., 2013; Zarzoso-Lacoste et al., 2016).

Amphibians represented 0.85% of the total items ingested by the otters and were included in the diet of pools from the upper part of the river. This finding could depend on the ecological requirements of the autochthonous Rana italica, endemic of the Italian peninsula, living in the source of mountain streams, linked to clear and cool water. (Smiroldo et al., 2009) reported that the frequency of consumption of amphibians in the diet of the otter increased with the altitude of sampling stations (Remonti et al., 2008; Smiroldo et al., 2009).

In some cases, molecular analyses allowed to resolve uncertainties emerging from the morphological approach. For example, morphological analyses revealed the presence of Rana sp. and Bufo sp. (Prigioni et al., 2006a; Smiroldo et al., 2009) while molecular analyses refined this assignment to Rana italica and Bufo bufo, respectively. The diversity of fish in the otter diet shows an impoverishment in the pools sampled upstream of the dam. Also, from the multivariate analysis (biplot) pool 5 seems to be characterized by a greater number of variables. This trend could be consistent with the impact of these anthropogenic structures on fish diversity, probably hindering the movements of some species along the river axis. In some cases, the impacts of the dam were compensated by building channels for lifts that bypass the barrier (Harris et al., 2017; Voicu et al., 2020).

Conclusion

Our study effectively highlights the variability in the diet of closely related otter populations confirming some evidence put forth by other authors, and allowing the depiction of a more precise trophic niche for this species. Indigenous species of cyprinids represent the main trophic source, although alien species constitute a non-negligible percentage of their diet.

In fact, out of twelve alien species only three (Carassius sp., Cyprinus carpio and Perca fluviatilis) stand out with an appreciable presence, and this occurs primarily close to lake environments (Lanszki et al., 2007). The recent review on the importance of non-native fish in the diet suggests their slight increase with time, probably as a consequence of alterations in the fish assemblages (Balestrieri, Remonti & Prigioni, 2016) . Could this also be the case of Pertusillo lake and Agri system in general? Probably yes, if we consider that the introduction of alien species has been growing over time, as highlighted also by the presence of Aspius aspius, a species that has only recently found much appreciation among anglers. This data seems to be alarming, if we consider that, in the near future the fresh waters of Basilicata will be populated by allochthonous species up to 78% (Caricato, Canitano & Montemurro, 2014).

Our findings could help define strategies useful for the conservation of the otter in Southern Italy, suggesting management actions directed at fishing regulations affecting food availability. In order to avoid alterations of otter food availability and native fish communities, some actions need to be taken. For example, the interruption of all kinds of introduction in the National Park and adjoining areas, especially using stocks of unknow origin; as well as controls on the health and taxonomic status of stocks using genetic approaches.

Supplemental Information

Supplemental Information 1 Primer validation and data availability

Click here for additional data file.

We thanks to Ente Parco Nazionale dell’Appennino Lucano, Val d’Agri-Lagonegrese for supplying some samples. In particular, we are grateful to Remo Bartolomei, Donata Coppola, and Luciano Ferraro for their help during sample collection. We are grateful to Gianluca Zuffi for valuable suggestions provided about the ichthyology of Italy.

Additional Information and Declarations

Competing Interests

Author Contributions

Field Study Permissions

Data Availability

The authors declare there are no competing interests.

Maria Buglione and Domenico Fulgione conceived and designed the experiments, performed the experiments, analyzed the data, prepared figures and/or tables, authored or reviewed drafts of the paper, and approved the final draft.

Simona Petrelli and Claudia Troiano performed the experiments, analyzed the data, prepared figures and/or tables, authored or reviewed drafts of the paper, and approved the final draft.

Tommaso Notomista and Eleonora Rivieccio performed the experiments, prepared figures and/or tables, and approved the final draft.

The following information was supplied relating to field study approvals (i.e., approving body and any reference numbers):

The collection of excrements at the study site did not need permission by the managing institution of the protected area. However, the activities were developed with the scope of a project authorized by the Ente Parco Nazionale dell’Appennino Lucano, Val d’Agri-Lagonegrese.

The following information was supplied regarding data availability:

Raw sequencing data are available in the ENA’s Sequence Read Archive: PRJEB38720 and at FigShare: Fulgione, Domenico; Buglione, Maria; Petrelli, Simona; Troiano, Claudia; Notomista, Tommaso; Rivieccio, Eleonora (2020): Fastq-20200326T163421Z-001.zip. figshare. Dataset. 10.6084/m9.figshare.12034920.v1

10.6084/m9.figshare.12034920.v1 https://www.ebi.ac.uk/ena/browser/view/PRJEB38720.

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
