# Peer review of "The diet of otters (Lutra lutra) on the Agri river system, one of the most important presence sites in Italy: a molecular approach"

_PeerJ, doi:10.7717/peerj.9606_

## Round 0.1 · original submission · Major Revisions

Both reviewers appreciated the work described in this paper, but they also point to some major issues. I would like to emphasize one in particular - the "novelty" aspect (i.e. being the "first" one etc.). This is definitely not needed (and anyway if you narrow down the focus enough you are always the first ones!). What needs to be emphasized is what makes the study a high-quality study - with respect to methods, and /or that the questions asked (the "why" of the study, an aspect one reviewer was requiring) are answered unambiguously.

Reviewer 1 ·

Basic reporting

Clear and unambiguous, professional English used throughout.

The majority of the manuscript is well written und easy to understand, however, the authors fail to define some of the metrics they use such as „frequency of occurrence“ or „relative presence“ these might not be common knowledge for all potential readers. Additionally, the English is sometimes flawed. I tried to point out some errors and minor comments below, but would advise the authors to seek the help of a native speaker for proof reading.
Lines 64-65: “please rephrase “deserving of conservation actions”
Line 69: please rephrase “basic for”
Line 105: please rephrase “full of”
Line 113: please rephrase “aged”
Line 114: please remove “been”
Line 116: 0°C is a very unusual temperature for preservation, please explain
Line 129: please add “gene” at the end of the heading
Line 145: how were the 50ng of DNA measured? Was 3each sample run on a Qbit or nanodrop before? Please explain.
Line 152: please rephrase “amplification” (probably to plate?)
Line 154: please explain if the full 25µl were combined and replace “in” with “of”
Line 157: please consider replacing “hunting booty” with prey (spectrum)
Line 159: please rephrase “flowing upstream”
Lines 182-183: please rephrase “making”
Lines 208-209: Please explain in detail how the negative controls were handled during and after sequencing regarding the equimolar pooling and how it was determined, that they do not contain contamination.
Line 220: please rephrase “but there are not necessarily these”
Line 239: please rephrase “most indigenous species”


Literature references, sufficient field background/context provided.

Regarding the cited references I have two major concerns: First, the literature references regarding the molecular analyses seem a bit dated and I would strongly advise the authors to have a look at Deagle et al. 2019 (Counting with DNA in metabarcoding studies: How should we convert sequence reads to dietary data?) as it not only provides a great literature summary, but also input for improvements of the analysis of the present manuscript. Additionally, the used statistics software was not named.
Second, the main focus of this manuscript is otter diet. However, the introduction primarily focuses on otter distribution and contains only little information on the available prey species in the study region. It would be a great improvement of the manuscript if the reader is presented with this information in the introduction instead of learning about it bit by bit in the results and discussion.
A third concern is, that the authors do not reference other (molecular) Eurasian otter studies carried out in other countries (e.g. Hong et al 2019, Lanski et al 2008, 2010 or Carss et al 1995, 1996)) and do not provide information on the otter population in the study area. An approximate individual count or territory count would be beneficial to the present manuscript.


Professional article structure, figures, tables. Raw data shared.

I found the structure of the article conform to the usual standards. However, there is an imbalance between the different sections: whilst the introduction is missing critical background information, parts of the Materials and Methods could be omitted. This concerns the description of the local flora and the river structure outside of the study area (lines 90-97). Additionally, the last part of the result section is rather repetitive and could be condensed considerably. The Materials and Methods section does not have a heading but is otherwise informative and well written.
The figures are well connected to the manuscript, but their resolution is rather low. In Fig. 1 the Caolo river is not named. For Fig. 2 and 3 a lot of information is missing in the caption and in Fig. 3 some text is overlapping with a line.
Regarding the Tables, I would suggest to either remove Table 1 and include the information on spraint counts in the text or remove the respective paragraph in the main text and add this information to the figure. The Caption of Table2 should include definitions of the used abbreviations.
I found the raw data accessible and sufficiently described.


Self-contained with relevant results to hypotheses.

The fact that the authors include a sentence on the amplification of invertebrate DNA from the samples in the abstract and this is never mentioned in the main text of the manuscript has me doubting if the manuscript is self-contained. Unfortunately, the authors do also not state any hypotheses. Nevertheless, I do think the work represents an appropriate “unit of publication”.

Experimental design

Original primary research within Aims and Scope of the journal.

The current manuscript fits well into the Scope of the Journal.


Research question well defined, relevant & meaningful. It is stated how research fills an identified knowledge gap.

The authors clearly state the usefulness of the presented work and that the molecular analysis of otter spraints can help with the clarification of prey species identity of taxa which are not easy to identify with morphological methods. As a reviewer is was, however difficult to assess if and how the study contributed to adding to the knowledge on otter diet in the study area, as the manuscript at the moment does not contain a description of the available prey items, and species, which are difficult to distinguish morphologically.


Rigorous investigation performed to a high technical & ethical standard.

There is one aspect of the present manuscript for which the presented information is not sufficient to determine of this criterion was met: the results of the NCBI blast search matching the obtained sequences to species or higher taxonomic entities. The Blast search is a key aspect for the whole manuscript and the validity of the results. Without knowledge on
• the locally available prey species,
• how many of them have 16S sequences published on Genbank,
• the suitability of the chosen target fragment to distinguish between all of them
it is impossible to judge the quality of the presented results. I would strongly suggest the authors to add a list of potential prey species and an alignment of their 16S sequences to the paper. This could also explain why certain sequences can only be assigned to genus or family. In case most of the local prey species have publicly available 16S sequences, blasting against a custom and well curated reference sequence database might improve the taxonomic resolution of the results substantially.

Additionally, the use of read counts as a measure of ingested biomass remains highly controversial in the field (e.g. Deagle et al 2019) and certain prerequisites should be fulfilled before doing so (e.g. primer matches equally well to all targeted species). It would be good to include some of this background information in the manuscript or put more focus on prey information obtained at the spraint level. Especially since the authors seem to use read counts as an equivalent to ingested prey biomass across different taxa (e.g. sentence in abstract) and a multitude of studies to this date show, that this is often not the case (e.g. Thomas et al 2014 and 2015).


Methods described with sufficient detail & information to replicate.

The description of the used methods is well comprehensible. For clarification purpose it should be mentioned if the morphological analysis was done before or after DNA extraction and if whole or parts of the spraints were extracted and how the laboratory was organized regarding measures against contamination.

Validity of the findings

Impact and novelty not assessed. Negative/inconclusive results accepted. Meaningful replication encouraged where rationale & benefit to literature is clearly stated.

As summarized above, the presented study shows enough levels of novelty and importance. To ensure the validity of the findings, I suggest adding the information proposed above. In my opinion this is crucial for this manuscript as the authors stress that this is not a proof of concept study, but that the data should inform conservation and management measures in the future.


All underlying data have been provided; they are robust, statistically sound, & controlled.

The raw data are available on Figshare and the step by step processing of the reads is sufficiently described.


Conclusions are well stated, linked to original research question & limited to supporting results.

The presented conclusions fit to the obtained results; however, several aspects of the paper warrant additional discussion: for instance, the size of the fish prey, life stage of the otter, and prey abundance could have a large influence on the frequency with which a certain taxon is consumed. Depending on the number of otters present at each of the five sampling sites, the detected prey taxa could even indicate individual preferences. Therefore, the authors should discuss the influence of elevation above sea level onto the consumption of amphibians with caution. In case morphological analysis did not only target crustacean remains, these data could be useful for the interpretation of the obtained molecular results (see papers by Carss et al.).

Additional comments

The manuscript shows nicely, how non-invasive sampling and the identification of prey species can help with the conservation and management of endangered species and it was an enjoyable read. Unfortunately, I can not recommend it for publication in its present state because the authors do not provide all necessary information to confirm the validity of their results and do not include crucial background information on the study system, related studies, and the used statistical metrics.

Reviewer 2 ·

Basic reporting

Language quality is OK.
Literature references must be updated: there are two prior papers on the subject, while the authors claim to be the first ones.
The manuscript needs a better focus. A mere description of the diet of otters does not suffice. Actually, I think that changing the angle of the ms having a hypothesis/question to be addressed by knowledge of the diet of otters will help the authors to write a better structured discussion (and conclusions).

See suggestions in section: General comments for the authors.

Experimental design

Study within the aims and scope of the journal.
Clarification is needed with regard to the selection of primers, absence of mock communities and if chimera removal was performed or not.
I am not expert in the bioinformatic analysis, so I cannot evaluate if methods are described with enough detail to be replicated, but my feeling is that they are.

See suggestions in section: General comments for the authors.

Validity of the findings

In the preset state, I would not consider the ms as novel. Should the authors change the focus, it might increase impact dramatically!
No problem with data.
Conclusions need to be rewritten.
No problem with speculation.

See suggestions in section: General comments for the authors.

Additional comments

The manuscript by Buglione et al. describes the diet of the Eurasian otter (Lutra lutra) in ITaly, where the species is endangered. The focus of this research is on the use of DNA metabarcoding.

Unfortunatly, I cannot recommend the publication of this manuscript in its current form, but I hope that the following comments will be useful to improve the work.

1. Lines 1 an 28. This is not the first use of DNA metabarcoding to study the diet of otters. Kumari et al. (2019) and Martínez-Abraín et al. (2020) used DNA metabarcoding with spraints of Lutra lutra. THe former worked with otters from South Korea and “targeted 12S rRNA gene region for vertebrates, 16S rRNA gene region for invertebrates, and cytochrome c oxidase 1 (COI) gene region for fishes”. The latter used a fragment of the 12S rRNA as it focused on fish diet of otters inhabiting a Spanish reservoir.
2. Abstract: line 20. Consider using “in the area between Naples and the Ionian Sea” as many readers will be unfamiliar with the terms “Basilicata” and “Campania”.
3. Line 34: change “analyzes” to “analyses”.
4. LInes 35-40. It is not clear why you write here about the crustaceans and then about the additional universal set of primers. From the previous lines, I had understood that your focus was on the vertebrates… so these lines here are puzzling. Actually, as the introduction of your abstract deals with conservation, I would expect here a couple of lines describing the ecological/conservation implications of your work. Current lines 40-42 are a mere wish, so I suggest you to link your results with putative actions to be undertaken. See comment number 7 below.
5. Introduction: please show the current distribution of the species in Italy by means of a map.
6. Line 64. Please elaborate on the official category of threat at national and regional level.
7. Lines 66-74. The authors should illustrate the use of knowledge about the diet of composition for conservation purposes with at least one example. The current writing seems a bit a vague with regard to the usefulness of their work. You need to better justify your investigation on the diet of otters in Campania. A mere description is not enough for a journal such as PEERJ. It could be that you expect a different diet because of whatever reason, or you might be intrigued by a putative competition between otters and anglers for the same species, or might even investigate the role of native versus exotic species in the diet of your otters…
8. Lines 130-134. It could be useful for readers unfamiliar with the ecology of otters in the study area to specify that low importance of crustaceans. For instance, Procambarus clarkii is really important for the diet of Iberian otters.
9. Section Sample collection: is there any record of how many individuals may form the population for each sampled area?
10. Section Sample collection. Please indicate here that you collected 51 samples. Readers will have to wait otherwise until line 199 as one cannot precisely tell from Figure 1.
11. Lines 146-148. Please write the PCR recipe indicating the final concentration of each reagent, rather that indicating the volume taken from the concentration of the product you purchased.
12. At least two mock communities should have been analysed. Mock communities are highly recommended to calibrate the filtering settings to be applied during the bioinformatic analysis in order to eliminate erroneous OTU assignments (Bokulich et al. 2013). This is especially important as you write in lines 202-204 that the scope of your ms is to introduce genetic analysis as a method for developing qualitative and quantitative comparisons of the otter diet
13. More about quantitative comparisons. You write in lines 237-240 that “the average percentage presence of the different animal species in the food of the otter…” which is a misleading sentence. As you are perfectly aware of, OTU relative abundance is defined as the number of reads assigned to that OTU divided by the total number of reads. At sample level, it might be that those “highly represented species” simply amplified better. For instance, if SPECIES A is represented by the 30 % of the sequences in spraint 1 and SPECIES B is represented by the 60 % of the sequences in the same spraint, we cannot reliably conclude that there was more SPECIES B DNA in spraint 1. Having said this, WITHIN THE SAME STUDY, the PCR bias should go in the same direction, so you may compare how the abundance of a given taxon changes. For example, if SPECIES A is represented by the 40 % of the sequences in spraint 1 and by the 15 % in spraint 2, we can conclude that there was less SPECIES A DNA in spraint 2 (Schloss et al. 2012; Geisen et al. 2015, Matesanz et al. 2019). I think this kind of explanation is needed in a journal such as PEERJ, where many readers will be not experts in metabarcoding.
14. Have I missed how did you detect and remove chimeras from your data? Please, clarify. IF you haven’t done it, the bioinformatic analysis should be repeated including this step.
15. Lines 216-223. If your focus was simply on Pisces and Amphibia, wouldn’t be better to have designed primers specific for these two taxonomic groups? Messing around with mammal and bird sequences does not seem a very efficient approach, does it not? I see that specific primers for fish and amphibians exist (Ac16S or the L2513/H2714 primer sets, Evans et al. 2015), so please justify the use of primers 16s mam1 and 16s mam2 in your work.
16. Lines 247-252. Please quantify the species richness and provide statistical support for the statement of change of diet of otters according to the structure of the watercourse. You might take a look to the approach taken by Martínez-Abraín et al. (2020).
17. Unfortunately, I cannot praise the Discussion of the current manuscript. In my opinion, it is redundant with some issues explained in Results, and difficult to read. I am sure that if you change the focus of the manuscript (from a mere description of diet to how that description solves a biological question), the discussion will be much easier to write. I also recommend to use sub-sections so that the reader goes from topic to topic smoothly. Your finding of species not previously recorded in the area (Ballerus sapa, Aspius aspius and Blicca bjoerkna) and the lowest diversity in the lower reach of the river with the dam as threshold are extremely interesting. But I would not spend so many lines emphasising that you are testing the method: DNA metabarcoding is well known to have better taxonomic precision than morphological analyses and it is already a well-established methodology, with two published papers in otters.
18. The section CONCLUSIONS has to be completely rewritten as well. Again, once a proper focus is addressed, it will be easier to write. Just an example, the use of blocking primers was never mentioned before (nor the problem of not using them…) and appears in these very last lines. The conclusion needs first a proper question to be answered; find it and you will have a fine ms.
19. Lines 439. It should read “Acknowledgments”.
20. able 3: use only two decimals.
21. All through the manuscript: when writing a genus name (e.g. Carassius) and “sp.”, “sp.” Should not go in italics.
22. Line 383. It should read “prove”.


References:
Bokulich et al. (2013) DOI: 10.1038/nmeth.2276.
Evans et al. (2015) DOI: 10.1111/1755-0998.12433
Geisen et al. (2015) https://doi.org/10.1111/mec.13238
Kumari et al. (2019) https://doi.org/10.1371/journal.pone.0226253
Martínez-Abraín et al. (2020) https://doi.org/10.1007/s10750-020-04208-y
Matesanz et al. (2019) https://doi.org/10.1111/1755-0998.13049
Schloss et al. (2011) https://doi.org/10.1371/journal.pone.0027310

---

## Round 0.2 · Major Revisions

I have now read your revised paper together with your detailed response to reviewers' comments. You have answered well most of these comments, but there are two comments that have not been adequately answered: 1) Reviewer 1 wrote that the English was sometimes flawed, you wrote that you had proof-read the manuscript and "delate" (delete) other language inaccuracies. I cannot believe that you have asked a native reader (as Rev 1 suggested) to proof read the manuscript as it is still poorly written in many places (eg just in results section in the summary "represented 0.85%" (same thing l. 273), "range of available prey" (not prey availability), "severe taxonomic control" does not really mean anything, the same with "homogenization of otter preys". You need to check the language throughout the paper (as other examples that you have not really proofread carefully the paper, some scientific names are with a capital letter for species: l. 120 Esox lucius, l. 131 and 431 Bufo bufo, l. 120 Rutilus is without species name, is it rubilio?). 2) The statistical analysis section that you wrote needs to be corrected: use percent (not per cent), no need to refer to boxplot using R (what is needed is to describe what is shown in a boxplot - ie median, quartiles, extreme values), it is alpha diversity, not alfa, you need to define what kind of Chisquare values you have calculated, and finally there is no such thing as a two-dimensional PCA. You can use as many axes as needed in a PCA (it depends on how much variation you have on different axes), since it is a method used to summarize and describe multivariate patterns. I would not recommend using PCA on such data as Correspondence Analysis is often best at analysing such occurrence tables (eg Borcard et al. 2011), and is easily implemented in R.

l. 78-79 - I don't think you mean that the number of individuals vary between 25 and 27 individuals ("oscillating") - I would just write that there are approximately 25 individuals in the study area.

I would also modify the first sentence of the discussion about otter as keystone predator - studies by J Estes for example are on sea otter in the Pacific and clearly cannot be used in this context (and the reviews by Pace or the book by Begon refer to these studies). This is a misuse of the literature. (River) otter might play a significant role, but you do not provide evidence for it.

Borcard, D., Gillet, F., & Legendre, P. (2011). Numerical Ecology with R New York: Springer.

---

## Round 0.3 · accepted · Accept

The language has been improved - note that formally, a test does not measure the likelihood that observed differences might be due to chance, but it is the probability to get data under the null hypothesis of identical proportions (in your case of analysing contingency tables) that are as AND MORE extreme than the observed sample, extreme being defined by the test statistic.